# Aryl Hydrocarbon Receptors: Evidence of Therapeutic Targets in Chronic Inflammatory Skin Diseases

**DOI:** 10.3390/biomedicines10051087

**Published:** 2022-05-07

**Authors:** Han-Bi Kim, Ji-Young Um, Bo-Young Chung, Jin-Cheol Kim, Seok-Young Kang, Chun-Wook Park, Hye-One Kim

**Affiliations:** Department of Dermatology, College of Medicine, Hallym University, Kangnam Sacred Heart Hospital, Seoul 150-950, Korea; khmamy1029@naver.com (H.-B.K.); ujy0402@hanmail.net (J.-Y.U.); victoryby@naver.com (B.-Y.C.); aiekfne@naver.com (J.-C.K.); tjdjrdud@naver.com (S.-Y.K.); dermap@hanmail.net (C.-W.P.)

**Keywords:** aryl hydrocarbon receptor, CYP1A1, psoriasis, TCDD, 2,3,7,8-tetrachlorodibenzo-p-dioxin

## Abstract

The aryl hydrocarbon receptor (AhR), a ligand-dependent transcription factor, is important for xenobiotic metabolism and binds to various endogenous and exogenous ligands present in the skin. AhR is known to be associated with diseases in various organs; however, its functions in chronic inflammatory skin diseases, such as atopic dermatitis (AD) and psoriasis (PS), have recently been elucidated. Here, we discuss the molecular mechanisms of AhR related to chronic inflammatory skin diseases, such as AD and PS, and the mechanisms of action of AhR on the skin immune system. The importance of AhR molecular biological pathways, clinical features in animal models, and AhR ligands in skin diseases need to be investigated. In conclusion, the therapeutic effects of AhR ligands are demonstrated based on the relationship between AhR and skin diseases. Nevertheless, further studies are required to elucidate the detailed roles of AhR in chronic inflammatory skin diseases.

## 1. Introduction

The aryl hydrocarbon receptor (AhR), which binds to exogenous and endogenous chemicals and regulates the expression of several genes, is a ligand-dependent transcription factor. It is ubiquitously expressed in various cells, including those of the skin derived from early evolutionary organisms, and exhibits either a positive or negative effect [1].

AhR is a sensor for xenobiotic chemicals that modulates adaptive and toxic responses to a variety of chemical contaminants, including polycyclic aromatic hydrocarbons and polychlorinated dioxins, especially 2,3,7,8-tetrachlorodibenzo-p-dioxin (TCDD) [2]. It also acts as a modulator of enzymes such as cytochrome P450 that metabolize these chemicals. According to a previous study, AhR plays a multifaceted physiological role as an environmental, dietary, or microbial signal [2]. For instance, the action of AhR in liver and breast inflammation has been examined [2]. The role of AhR in various immune responses is also crucial [3].

AhR has been shown to have important functions in the differentiation of many developmental processes, including those related to hematopoiesis, the lymphatic system, T cells, neurons, hepatocytes, and hematopoietic stem cells. AhR has recently been recognized as a crucial modulator of host-environment interactions in immune and inflammatory responses [3,4,5]. AhR is highly expressed in all skin cell types and regulates many genes that are important for basic skin function [6]. AhR signaling plays an important role in the development of skin barrier and melanogenesis in human skin cells [7]. Furthermore, AhR signaling is critically involved in the pathogenesis of several diseases involving organs such as the lungs [8], kidneys, liver [2], breast [6], and central nervous system [9]. The role of the AhR signaling pathway in cardiac toxicity associated with acute lead poisoning has been studied under in vitro and in vivo conditions in murine models [9]. Therefore, *AHR* may be a potential therapeutic target gene. In recent years, interest in the effects of AhR on psoriasis (PS) and atopic dermatitis (AD) has steadily increased. It started with the discovery that exogenous AhR ligands related to activities such as smoking and air pollution exacerbate these two chronic inflammatory skin diseases. AhR is involved in the regulation of skin immunity and barrier function through a complex mechanism rather than simply exacerbating these inflammatory diseases. These factors may play conflicting roles in inflammation. Keeping this in mind, the purpose of this review is to organize the research outputs related to AhR and inflammatory diseases. Towards this end, we review in vivo and in vitro studies on AhR, primarily focusing on psoriasis and atopic dermatitis.

## 2. Molecular Mechanisms of AhR Pathway in the Skin

### 2.1. Canonical

In a recent study, AhR was found to regulate canonical and non-canonical pathways (Figure 1). The canonical signaling pathways are described as follows. First, AhR canonical signaling begins in the cytoplasm, wherein AhR is bound by a chaperone complex [10]. AhR regulates xenobiotic-metabolizing enzymes, such as cytochrome P450 1A1 (CYP1A1), which are expressed widely throughout the human body [11]. In the canonical AhR signaling pathway, CYP1A1 is induced by AhR activation [12,13,14]. AhR exists in the cytoplasm as a multi-protein complex consisting of two chaperone proteins [15]. Activated by a ligand, the AhR complex eventually binds to a DNA-recognized genomic region, the dioxin response element (DRE) located upstream of the *CYP1A1* and *AhR repressor* (*AhRR*) genes (Figure 1) [16]. Induction of transcription of additional phase 1 xenobiotic metabolizing enzymes such as *CYP1A2* and *CYP1A1* is driven by the AhR complex signaling pathway [17,18]. It regulates AhR activation by three different checkpoints in the canonical AhR signaling pathway: (a) proteasomal degradation of AhR, (b) metabolism of ligands by CYP1A1, and (c) disruption of the AhR:ARNT complex by AhRR [17,18,19]. The role of aryl hydrocarbon receptor nuclear translocation proteins in the action of aryl hydrocarbon (dioxin) receptor is also known. The Aryl hydrocarbon receptor nuclear translocator (Arnt) is a basic helix-loop-helix transcription factor that heterodimerizes with AhR to mediate signal transduction pathways stimulated by 2,3,7,8-tetrachlorodibenzo-p-dioxin [19]. In addition, it may result in activation of the tyrosine kinase c-Src by ligand-induced dissociation of the cytoplasmic multiprotein complex, which in turn activates epidermal growth factor receptor and downstream mitogen-activated protein kinase (MAPK) signaling [19]. TCDD is one of the ligands that mediate signal transduction through the protein phosphorylation pathway and c-Src, an essential component of the cytoplasmic AhR complex [19].

### 2.2. Non-Canonical

AhR is activated in the absence of DRE at different transcription start sites in AhR-responsive genes, suggesting a non-canonical pathway (Figure 1) [20,21]. A direct interaction between AhR and nuclear factor-κB (NF-κB) decreases the expression of CYP1A1 and induces the expression of cytokines and chemokines, such as the B cell-activating factor of the tumor necrosis factor family and the transcription factor interferon responsive factor [22].

The involvement of RelB in AhR-mediated induction of chemokines is also known [22]. This was also confirmed by the results of our study on AhR and NF-kapaB interaction at the epigenetic stage. Our published data indicates that treatment with Bay-117082, an inhibitor of NF-κB activation, reduces the activation of AhR-related genes (e.g., *CYP1A1* and *AhRR*) [23]. 

Research on AhR often leads to contradictory interpretations of inflammatory skin diseases. Recent studies have shown that blocking AhR activation is desirable in some skin conditions; however, in the opposite case, stimulating AhR activation inhibits skin inflammation. In the following section, we examine the conflicting findings.

## 3. AhR Is an Exacerbation Factor for Inflammatory Skin Disease

### 3.1. Cutaneous Immune Function

AhR regulates cell differentiation and plasticity, and its overactivation causes severe skin lesions in humans [24]. Dioxin, one of the ligands of AhR, has been reported to induce skin inflammation in vivo. In an in vivo study, Rudyak et al. reported that TCDD treatment alone increased dermal infiltration by mast cells, macrophages and several inflammatory cells in HRS/h/hairless mice. Induction of an increase in inflammatory expression by TCDD was also demonstrated. In addition, increased expression of cytokines such as inflammatory markers IL1β, IL6, IL22, TNF-α and cysteine-rich protein 61 was also confirmed [25].

### 3.2. In Vivo Studies on AhR

To date, many scientists have examined the effects of exposure to TCDD (an AhR agonist) in humans and the mechanisms of AhR activation. Takanori Hidaka et al. confirmed that epidermal keratinocyte-specific constitutive AhR activation is induced in atopic dermatitis (AD)-like inflammation [26]. Moreover, several investigators have attempted to confirm the previously observed role of TCDD in AhR metabolic enzyme activities, and AhR-null mouse models were created in the mid-1990s. In fact, the generation of such mice led to a full-blown transformation of studies on AhR [27]. An AhR-null mouse model was generated to characterize mice lacking AhR expression [28,29,30]. In general, AhR deficiency in non-hematopoietic cells exacerbates skin inflammation [31]. 

### 3.3. In Vitro AhR Assay

According to a previous study, AhR is involved in mediating the effects of antioxidant phytochemicals in atopic dermatitis [32]. AhR is a chemical sensor abundantly expressed in epidermal keratinocytes. Oxidative AhR ligands induce the production of reactive oxygen species (ROS) [32]. Upon ligand binding, cytosolic AhR undergoes nuclear translocation, and transcription of various AhR-responsive genes is induced by CYP1A1. Although it detoxifies polyaromatic compounds, AhR can be detrimental because the activity of the CYP1A1 enzyme generates ROS in mutant metabolites [1,7,33,34]. It is the environmental pollutant benzo(a)pyrene that induces oxidative stress-mediated interleukin-8 production in human keratinocytes via the AhR signaling pathway [33,34].

## 4. AhR as a Mitigating Factor in Inflammatory Skin Disease

### 4.1. Cutaneous Immune and Barrier Function

In skin tissue, AhR plays an important role in maintaining skin homeostasis, such as environmental toxin metabolism, intracellular redox balance, response to ultraviolet (UV) light, diseases related to melanogenesis, regulation of immune processes and epidermal barrier function (Figure 2) [35]. Therefore, the relationship between AhR and skin homeostasis has been studied in the context of barrier physiology, immunology and toxicology [35]. Activation of the OVO-like 1 (OVOL1) transcription factor is initiated by the AhR:ARNT complex, which, in turn, enhances the expression of filaggrin (FLG) and loricrin (LOR), accelerates epidermal barrier formation in the epidermis, and differentiates keratinocytes [36]. Additionally, AhR-mediated activation of transcription factors such as Nrf2 induces cytoprotective antioxidant responses that suppress oxidative stress and restore skin homeostasis (Figure 3 and Figure 4) [37,38]. 

Studies have also provided insights into the potential the role of tryptophan-derived AhR ligands by physiological and pathological processes in the skin. AhR-mediated immune response may be associated with high AhR expression in skin cells [39]. AhR is expressed on immune cells including antigen-presenting cells, T cells, fibroblasts, macrophages, mast cells and other skin immune cells.; The function of antigen-presenting cells, including Langerhans cells, and cytokine expression are required by the expression of AhR [27]. Terminal differentiation of CD4+ T helper (Th) 17 and Th 22 cells and expression of IL-17 and IL-22 cytokines have been shown to be regulated by AhR signaling [40,41]. AhR is known to regulate the peptidoglycan-induced expression of inflammatory genes in human keratinocytes. Bacterial peptidoglycan-induced skin inflammation in keratinocytes is regulated by AhR, which modulates the expression of inflammatory genes [42]. In brief, AhR is involved in many skin functions, including those of the skin immune network as well as in cell homeostasis (Figure 3). 

### 4.2. In Vivo Studies on AhR

In one study, the dorsal skin of AhR-null mice showed hyperkeratosis, acanthosis, and marked dermal fibrosis, suggesting a role for AhR in controlling skin differentiation [43]. Consequently, skin wounds of AhR-null mice healed faster, probably because of reduced inflammation [44]. Paola Di Meglio et al. confirmed that AhR agonist 6-formylindolo[3,2-b] carbazole attenuates imiquimod-induced psoriasis (PS)-like dermatitis [31]. Several mice experiments were also investigated to investigate AhR function. Andreola F et al. confirmed that AhR-deficient mice are defective in retinoic acid metabolism [45]. Resistance to benzo[a]pyrene-induced carcinogens was confirmed [46]. Bettina Jux et al. suggest that AhR-deficient mice have defective Langerhans cell (LC) maturation [47], and Stephanie Kadow et al. found AhR-deficient mice are lacking dendritic epidermal T cells [48]. Chien-Hui Hong et al. confirmed that selective AhR knockout in the epidermal LC causes the loss of LC and immune phenotype skewing [49]. In addition, the association between AhR and skin barrier function has been studied [44]. In experiments performed by Hass et al. conditionally AhR-deficient mouse lines exhibited AhR targets identified by many barrier-associated genes by analysis of weak intercellular connectivity and gene expression in the epidermis of keratinocytes. Kim et al. reported that rapamycin, a molecule that reduces the formation of ROS, alleviated TCDD-and imiquimod-induced psoriasis-like skin dermatitis via AhR and autophagy modulation [50]. In a study performed on mouse keratinocytes, Rico-Leo et al. used AhR−/− mice to investigate the effect of AhR by receptor depletion [24]. These findings support that AhR regulates skin regeneration and homeostasis by ensuring epidermal stem cell identity, thus highlighting this receptor as a potential target for the treatment of skin pathologies [51].

### 4.3. In Vitro AhR Assay

AD is characterized by increased expression of the type 2 cytokines IL-4 and IL-13; in particular, IL-13 has been highlighted as having pathological significance. This has been shown by accumulating evidence of skin barrier function regulated by competition between the AhR axis (barrier upregulation) and the IL-13/IL-4-JAK\STAT6/STAT3 axis (barrier downregulation). (Figure 4) [52]. Kim et al. discovered AhR antagonists that block TCDD-driven enzymatic activity [53]. They identified a novel compound, 2-methyl-2H-pyrazole-3-carboxylic acid (2-methyl-4-o-tolylazo-phenyl)-amide (CH-223191), which potently inhibited TCDD-induced AhR-dependent transcription. Currently, CH-223191 is used to block the binding of TCDD to AhR, which inhibits the TCDD-mediated nuclear translocation and DNA binding of AhR.

Studies on the regulation of FLG, LOR, and IVL expression mediated by AhR have also been reported. AhR activation restores FLG expression by OVOL1 in AD. IL-13/IL-4 has been shown in epidermal keratinocytes to bind to the heterodimeric IL-4Rα/IL-13Rα1 and activate the downstream JAK1/TYK2/JAK2 and STAT6/STAT3 axes [54]. Inhibition of AhR-mediated upregulation of FLG, LOR and IVL appears to correlate with activation of the IL-13/IL-4-JAK-STAT6/STAT3 axis [28,55,56]. On the other hand, inhibiting IL-13/IL-4 mediated STAT6 phosphorylation, and restoring IL-13/IL-4 mediated FLG reduction is achieved by activation of the AhR axis [57]. In addition, cytoplasmic-to-nuclear translocation of OVOL1 and inhibition of FLG and LOR expression are induced by activation of the IL-13/IL-4-JAK-STAT6/STAT3 axis (Figure 4) [28,56]. AhR has effects of inflammatory skin diseases, but there is still a long way to go before its practical against inflammatory diseases. Further studies are needed to better understand clinical and pathological correlations.

## 5. Ligands of AhR in Skin

AhR activation depends on a specific ligand, a structurally diverse spectrum of synthetic and environmental chemicals including dietary components, microbiota-derived factors, and endogenous tryptophan metabolites, as well as the interaction with specific co-modulators of gene transcription and/or other transcription factors [58,59]. AhR, at the intersection of these interaction signaling networks, has been reported to demonstrate therapeutic value [59]. The list of AhR ligands is impressive and strikingly diverse. Because of the abundance of AhR, it plays an important role in several physiological processes, including ligand pleiotropic function, AhR, xenobiotic metabolism, immune response, cell proliferation, differentiation and apoptosis [60,61].

In the skin, FLG expression in keratinocytes is dependent on AhR activity, by which AhR ligation leads to nuclear translocation of OVOL1 and subsequent FLG transcription [62]. The AhR-ARNT-FLG signaling pathway can be activated by rapidly metabolized AhR ligands such as indole-3-aldehyde (IAId) or 6-formylindolo[3,2-b] carbazole (FICZ), as well as by exogenous dioxins [63,64]. AhR ligands are formed from several sources, including exogenous and endogenous ligands (Figure 5).

### 5.1. Exogenous Environmental Pollutants

Environmental pollutants of AhR ligands include dioxins, polycyclic aromatic hydrocarbons (PAHs), and halogenated aromatic hydrocarbons [65,66,67,68]. For example, benzopyrene, the major PAH in smoke, recruits Langerhans cells and polarizes the Th2/17 response during epidermal protein sensitization via AhRs. The relationship between smoking and the clinical severity of PS has also been demonstrated [52]. The AhR ligand in environmental pollutants is 2,3,7,8-tetrachlorodibenzo-p-dioxin (TCDD) [69]. TCDD is a most toxic compounds and one of a family of isomers known chemically as dibenzo-p-dioxins. It has various systemic effects at a wide range of exposure concentration, such as developmental defects, cancer, wasting syndrome, hepatosteatosis, thymus involution, and dysregulation of immune responses, which are widely observed in different species [70,71,72,73,74]. Additionally, it binds to AhR and balances mucosal reactivity through interleukin-22 [74].

### 5.2. Particulate Matter (PM)

In addition, PM consisting of ions, organic compounds, metals and PAHs induce ROS formation and autophagy in human keratinocytes [75,76]. PM2.5 led to keratinocyte proliferation and differentiation via AhR activation as a result of the hyperkeratotic epidermis [57,77]. It was confirmed that AhR expression is increased in patients with chronic inflammatory skin disease [77]. In a recent study, Kim et al. showed that PM2.5-induced TNF-α and FLG deficiency through AhR activation pathways, inducing skin barrier dysfunction [78]. Furthermore, a recent study on PM2.5 suggested that it can induce melanogenesis in keratinocytes via AhR-MAPK signaling pathways [78]. Although our results have not yet been published, the regulation of PM2.5-induced inflammatory cytokines by AhRs has been confirmed.

### 5.3. Endogenous Ligands

Endogenous ligands include tryptophan metabolites such as FICZ, kynurenines, and ligands provided by commensal microbiota [74]. The tryptophan catabolites of the microbiota bind to AhRs and balance mucosal reactivity via interleukin-22 [74].

Tryptophan-derived AhR ligands, kynurenine (KYN), kynurenic acid (KYNA), and FICZ also regulate melanoma cell proliferation, cell cycle regulation, and apoptosis [79]. KYN, the main metabolite of tryptophan in mammals, is a direct precursor of KYNA, anthranilic acid, and 3-hydroxykynurenine. The expression of *KYNU*, which encodes an enzyme involved in tryptophan metabolism, is upregulated in psoriatic skin lesions [79]. KYN activates CYP1A1 and plays a more important role in AhR-dependent immunological responses than in the metabolism of xenobiotics [80]. Moreover, KYN participates in disease tolerance pathways and represents a link between tryptophan catabolism and the AhR signaling pathway through immunosuppressive mechanisms [81]. Even in vitiligo, L-tryptophan metabolism influences the immune response, ROS, and aryl hydrocarbon receptor-mediated immune response signaling.

### 5.4. Tryptophan Metabolites Generated by UV Irradiation

UV light, which is a typical source of physiological ligands, generates a high-affinity ligand, FICZ, from tryptophan. In addition, hydrogen peroxide in the skin of vitiligo patients can lead to the formation of FICZ [79,81]. FICZ is known to be effectively metabolized by CYP1A1, which is an important link in AHR–CYP1A1 feedback regulation [82]. In human keratinocytes, FICZ promotes wound healing in an AhR-independent manner, through extracellular signal-regulated kinase (ERK) signaling. [83].

### 5.5. Tryptophan Metabolites Are Produced by the Skin Microbiome

AhR ligands, such as bioproducts of microbiota, may permeate through the stratum corneum, epidermis, and skin appendages, such as hair follicles, sweat, and sebum glands, which serve as a “port d’entrée” for skin microbiota to colonize the deeper layers of the stratum corneum [83,84]. Host-microbial interactions are also mediated via AhRs in skin treatment applications [83]. Tryptophan, a major source of AhR ligand precursors, is converted into derivatives or by-products through the metabolic activity or biochemical reactions of the microbiota [85,86]. Some microbiota can metabolize tryptophan into indole derivatives, such as indole-3-acetaldehyde and indole acetic acid, which are AhR agonists [74,87,88,89,90]. Previous reports have demonstrated the importance of microbiota-derived indole in regulating homeostasis through AhR activation [91,92,93].

## 6. AhR in Skin Diseases

### 6.1. Atopic Dermatitis (AD)

AD is a common and heterogeneous eczematous skin disorder characterized by Th 2-deviated skin inflammation, barrier disruption, and chronic pruritus [94,95,96]. Skin barrier dysfunction is associated with reduced production of terminal differentiation molecules such as FLG [57,97]. During targeted ablation, a complex of AhRs in the mouse epidermis induces severe defects in ablation and epidermal barrier function [98]. The presence of a AhR protein and mRNA levels of *AhR* have been reported in AD [63,77]. Hong et al. has suggested increased AhR and ARNT expression without CYP1A1 induction in AD skin lesions compared to normal control skin [98]. Alternatively, Kim et al. has suggested increased expression of ARNT and CYP1A1 but not AhR in skin lesions of AD [77]. These findings collectively suggest that there is a lack of physiological ligands for AhR in the Th 2-prone milieu in AD. Air pollution contributes to the exacerbation and development of AD via the AhR pathway [99].

### 6.2. Psoriasis (PS)

The pathogenesis of increased skin infiltration and activation of effector CD4+ T cells, including upregulation of Th 17 and Th 22 cells, are hallmarks of PS [100,101]. It is characterized by the observation of unregulated AhR expression in psoriasis patients. Increased AhR expression levels in serum from peripheral blood monocytes and PS patients compared to healthy individuals ae associated with increased Th 22 cell and IL-22 expression [31,102]. Increased AhR expression in patients with chronic inflammatory skin disease has been confirmed in several studies [77]. Increased AhR expression has been demonstrated in skin biopsy samples of patients with PS, and treatment of skin cells with AhR ligands in vitro results in the modulation of genes, including IL-6, IL-8, and type I and II interferon pathway genes implicated in the pathogenesis of psoriasis [31,77]. Furthermore, in psoriasis, abnormal epidermal differentiation and impaired skin barrier function have been associated with the downregulation of the expression of skin barrier proteins such as FLG and LOR [103].

### 6.3. Vitiligo

Patients with vitiligo show significant upregulation of AhR transcription factor [103,104,105]. AhR functions in melanogenesis and melanocyte proliferation and differentiation and can modulate the production of cytokines such as IL-17a and IL-22 [103,104,105]. IL-17A and IL-22 are involved in vitiligo pathogenesis. In particular, AhR plays an important role in controlling the production of IL-22 in Th 22 cells [103,105]. Indeed, AhR expression was significantly lower in CD4+ T cells and the skin of vitiligo patients than that in healthy controls, and knockdown of AhR increased IL-17A production and decreased IL-22 levels in CD4+ T cells of vitiligo patients [31]. In contrast, there was a sharp increase in AhR mRNA expression in peripheral blood monocytes obtained from patients [105].

### 6.4. Acneiform Eruption/Chloracne

Chloracne is a condition characterized by acne-like eruptions of comedones, cysts, and pustules that develop after exposure to substances such as dioxins and dioxin-like compounds [106]. The pathogenesis of chloracne is also mediated by exposure to dangerous AhR ligands [106]. Substances such as dioxins and dioxin-like compounds increase the expression of the enzyme CYP1A1 in skin cells such as keratinocytes, sebum cells, and melanocytes of human skin through AhR, and overexpression of CYP1A1 has been confirmed in the skin of patients with chloracne [106,107]. The binding of dioxin to AhR accelerates epidermal terminal differentiation and converts sebaceous cells into keratinocytes, resulting in chloracne [107].

### 6.5. Hidradenitis Suppruativa (HS)

The link between pharyngitis AhR and possible fungal pathogenesis has been demonstrated [107]. HS is a debilitating, chronic, and recurrent skin disease of the hair follicles, which typically manifests as painful, deep-seated, inflamed lesions in the apocrine gland-bearing parts of the body; the etiology of the disease remains obscure [108]. However, it has been reported that IL-17 is strongly associated with the pathogenesis of HS (Hidradenitis suppruativa), and AhR is involved in the regulation of IL-17 secretion by Th 17 cells [108,109]. AhR primarily aids the differentiation of Th 17 cells in conjunction with exogenous or endogenous ligands. In addition, the skin microbiome regularly converts tryptophan derived from the host skin into indoles, which regulate tissue inflammation by binding to AhR [109]. Poorer AhR activation in HS skin lesions was confirmed, which coincided with reduced generation of bacteria-derived AhR agonists and a lower incidence of AhR ligand-producing bacteria in the local flora.

### 6.6. Skin Cancer

Tryptophan metabolites such as KYN, KYNA, and FICZ are AhR ligands that promote melanoma cell growth in vitro [110,111], have antiproliferative and cytotoxic activities, and promote apoptosis in melanoma A375 and RPMI7951 cells [110]. UVB, one of the major causes of melanoma, has been shown to enhance the antiproliferative activity of the KYN and KYNA, tryptophan metabolites of melanoma cells, A375, and SK-MEL-3 RPMI-7951 in vitro [111]. A potential biological interaction between UVB radiation and selected tryptophan-derived AhR ligands was also identified in melanoma cells. Moreover, it was confirmed that UVB increases the inhibitory activity of KYN and KYNA on the metabolic activity of melanoma SK-MEL-3 cells and enhances KYN-induced necrosis in these cells. However, the biological mode of action of each compound differs in different cells, and not all compounds are dependent on AhR [110,111].

## 7. Therapeutic Properties of AhR

The effects of AhR ligands as agonists or antagonists are dependent on several factors, including ligand structure, specific genes, and the cell context-dependent expression of important cofactors or coactivators [112]. The role of AhR in carcinogenesis has been examined, the effects of an AhR ligand antagonist have been explored, and the potential of AhR as a drug target has also been studied. As the Th 2-deviated milieu potently reduces FLG and other barrier-related molecules, the upregulation of the AhR:ARNT complex may compensate for attenuation of the Th 2-mediated FLG reduction in the skin barrier. A recent discovery related to endogenous AhR ligands revealed a physiological role for AhR in cell behavior and development, modulation of lymphoid cell development, and induction of regulatory T cells [113,114,115]. It is also AhR that regulates gut immunity through regulation of innate lymphocytes. The potential of a Th 2-biased environment to decrease the production of endogenous AhR ligands such as indole-3-aldehyde by symbiotic skin microbiota was studied by Yu et al. [115]. Therefore, FICZ and IAld, which are rapidly metabolized AhR ligands, may be helpful in the treatment of AD by appropriately activating the AhR-ARNT-FLG axis [62,116].

### 7.1. Coal Tar

Coal tar, which consists of a wide range of PAHs, is metabolized to detoxify internal PAHs via CYP450 enzymes induced by AhR [116,117,118,119]. Coal tar induces AhR-dependent skin barrier repair in atopic dermatitis. Recently, van den Bogaard et al. demonstrated that coal tar restored FLG expression in FLG-haploinsufficient keratinocytes and counteracted Th 2 cytokine-mediated downregulation of skin barrier proteins to wild-type levels. In addition, in AD patients, coal tar completely restored the expression of major skin barrier proteins, such as filaggrin, and diminished the levels of Th 2 cytokines IL-4 and IL-13 [57].

### 7.2. Oleanolic Acid

Oleanolic acid, a biological compound of *Lingustrum lucidum*, inhibits PM2.5-induced autophagy in keratinocytes. A study has been published on the regulation of oleanolic acid and AhR activation. In a study by Kim et al., oleanolic acid was shown to suppress CYP1A1 expression by AhR activation and reduce the levels of TNF-α and other inflammatory cytokines, such as IL-1β and IL-6 [120].

### 7.3. Tapinarof

Tapinarof (GSK2894512, previously known as WBI-1001) is a naturally derived small molecule produced by the bacterial symbionts of entomopathogenic nematodes [121,122]. Tapinarof is a natural AhR agonist used to treat skin inflammation in humans. Creams containing Tapinarof displayed significant efficacy in patients with PS and AD [123,124,125]. In a study by Smith et al., Tapinarof was shown to bind to and activate AhR in several other cell types, including cells of the human skin. In addition, it was reported that Tapinarof reduced the expression of pro-inflammatory cytokines in stimulated peripheral blood CD4+ T cells and ex vivo human skin and affected the expression of barrier genes in primary human keratinocytes. Compound actuation of erythema, epidermal thickening, and tissue cytokine levels was reduced by topical treatment of AhR-sufficient mice with Tapinarof in a mouse model of AhR [126]. Application of a Tapinarof 1% cream once daily showed superior results to those observed with vehicle control in reducing the severity of plaque PS [127]. Studies on Tapinarof 1% cream have reported long-term safety, efficacy and good tolerability in adolescents and adults with AD [128].

## 8. Conclusions

In summary, accumulating evidence suggests that various AhR pathways and ligands are involved in the pathogenesis of chronic inflammatory skin diseases, such as AD and PS. A therapeutic effect can be observed using AhR ligands based on the relationship between AhR and skin diseases. Nevertheless, the physiological and pathological actions and roles of AhR in the skin are still largely unknown, and further studies are needed in this area.

## Figures and Tables

**Figure 1 biomedicines-10-01087-f001:**
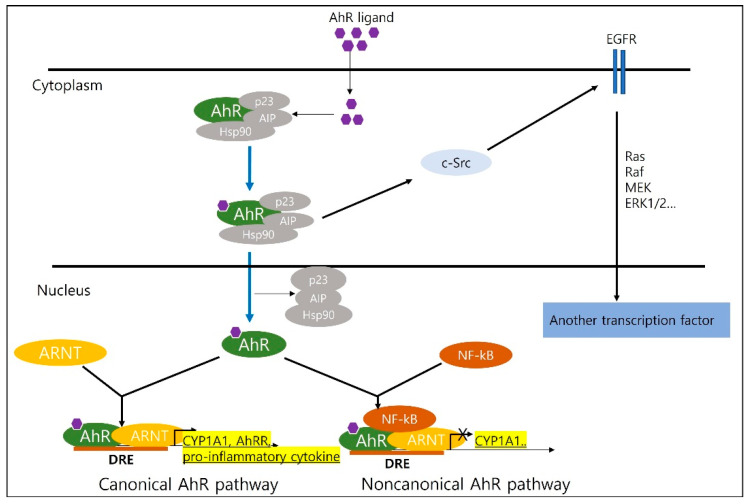
Canonical and non-canonical aryl hydrocarbon receptor (AhR) pathway. Canonical and non-canonical AhR pathway AhR regulates xenobiotic metabolizing enzymes such as cytochrome P4501A1 (CYP1A1), a canonical pathway, and activates in the absence of DRG at different transcription start sites of AhR responsive genes, suggesting a non-canonical pathway.

**Figure 2 biomedicines-10-01087-f002:**
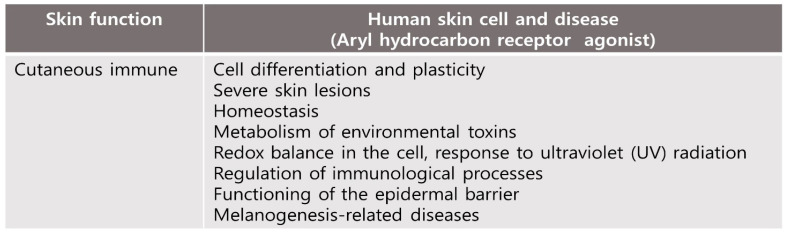
Effects of human skin cell and disease by TCDD (an AhR agonist) exposure and activation of AhR. AhR plays an important role in maintaining skin homeostasis in human skin.

**Figure 3 biomedicines-10-01087-f003:**
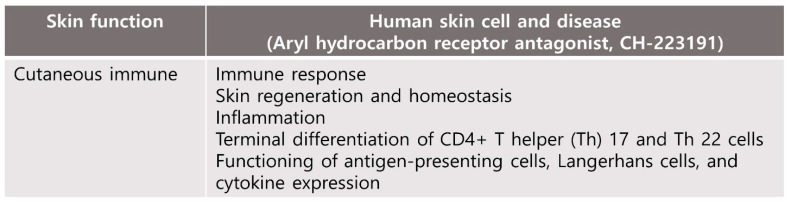
Effects of human skin cell and disease by AhR antagonist exposure and downregulation of AhR. AhR is involved in many skin functions including those of the skin immune network as well as in cell homeosis.

**Figure 4 biomedicines-10-01087-f004:**
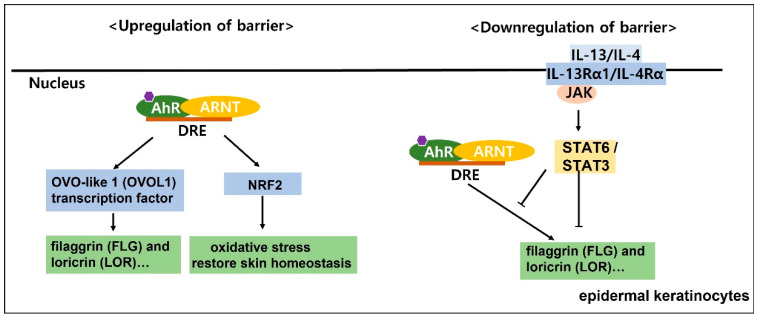
Skin barrier function is regulated by competition between the aryl hydrocarbon receptor (AhR) axis upregulation of barrier and downregulation of barrier. In epidermal keratinocytes, the AhR:ARNT complex initiates the activation of the OVO like 1(*OVOL1*) transcription factor, which subsequently enhances the expression of filaggrin (*FLG*) and loricrin (*LOR*) and accelerates the epidermis to form an epidermal barrier (upregulation of barrier). Activation of the IL-13/4-JAK-STAT3 axis inhibits the cytoplasmic to nuclear translocation of *OVOL1* and inhibits the expression of *FLG* and *LOR* (downregulation of barrier).

**Figure 5 biomedicines-10-01087-f005:**
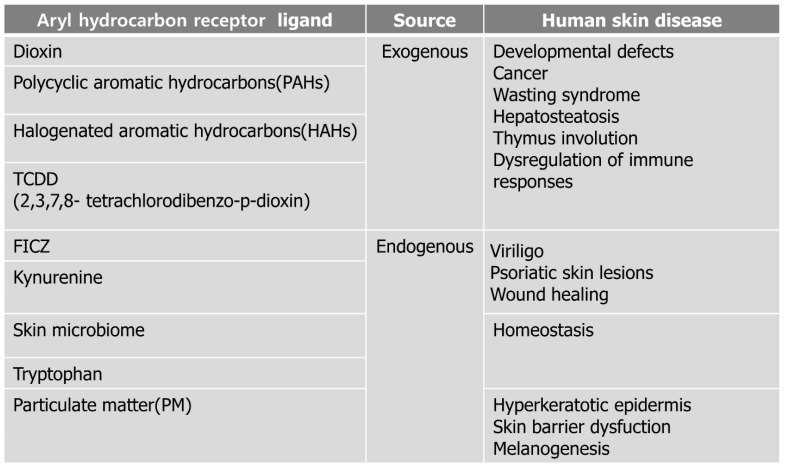
AhR ligands are classified according to exogenous and endogenous and also affect human disease.

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
