# Peer review of "Aryl Hydrocarbon Receptors: Evidence of Therapeutic Targets in Chronic Inflammatory Skin Diseases"

_biomedicines, 2022, doi:10.3390/biomedicines10051087_

Round 1

Reviewer 1 Report

This is a comprehensive  and timely review of the AhR and its ligands  as  potential therapeutic  targets in many   skin diseases such as AD, Psoriasis, HS, Acne, Vitiligo  etc.

The  IRB statements about healthy volunteers, and  5 men and 5 women with psoriasis seem to be unrelated to this review manuscript, and need to be removed. 

Author Response

[Reviewer 1]

This is a comprehensive and timely review of the AhR and its ligands as potential therapeutic targets in many   skin diseases such as AD, Psoriasis, HS, Acne, Vitiligo etc.

The IRB statements about healthy volunteers, and 5 men and 5 women with psoriasis seem to be unrelated to this review manuscript and need to be removed.

We would like to thank the Reviewer for the comment. deleted

Reviewer 2 Report

I appreciated the big effort of the AA. to evaluate and explain the importance of this very interesting and underestimated patway.Furthemore there is  a long way to reach the practical suggestions, at least, for the inflammatory disease.As far as clinical and pathological correlatons is concerned, I dont see, at moment,the great utility. However the theme discussed here is difficult and needs to be mentioned absolutely because we must open the mind to the "new molecules and understand their meaning and their use in always increasing pathology. To encourage the research, I would like underline one more time the frequence of diseases such psoriasis and atopic eczema  and their negative and devastating impact over the daily life of patients.

The paper is interesting and we are waiting for any suggestions useful for the therapy of inflammatory and chronic disease like Psoriasis and AD. 

Author Response

[Reviewer 2]

I appreciated the big effort of the AA. to evaluate and explain the importance of this very interesting and underestimated pathway. Furthemore there is a long way to reach the practical suggestions, at least, for the inflammatory disease.As far as clinical and pathological correlatons is concerned, I dont see, at moment, the great utility. However the theme discussed here is difficult and needs to be mentioned absolutely because we must open the mind to the "new molecules and understand their meaning and their use in always increasing pathology. To encourage the research, I would like underline one more time the frequence of diseases such psoriasis and atopic eczema and their negative and devastating impact over the daily life of patients.

We would like to thank the Reviewer for the comment. AhR has investigated the effects of inflammatory skin diseases, but there is still a long way to go before practical proposals for inflammatory diseases. Further studies are needed to better understand the clinical and pathological correlations.

The paper is interesting and we are waiting for any suggestions useful for the therapy of inflammatory and chronic disease like Psoriasis and AD. 

Reviewer 3 Report

Kim et al. reviewed the molecular mechanism of the aryl hydrocarbon receptor (AhR) for skin immunity.  

After briefly describing the modes of AhR pathway activation, i.e., canonical or non-canonical, they reviewed current understanding regarding the consequences of the activation, which can be either beneficial or detrimental. They also reviewed AhR ligands, either exogenous or endogenous, with disease-modifying effects.

Although this manuscript is well-written, the reviewer raises some concerns.

1. Chapters 3 and 4

   The reviewer strongly agrees with the manuscript arrangement since AhR activation can be therapeutic or disease-causing. However, some important studies regarding the roles of AhR in vivo are missing. For instance, AhR-deficient mice have defective retinoic acid metabolism (PMID: 9230184), resistant to benzo[a]pyrene-induced carcinogenesis (PMID: 10639156), defective Langerhans cell (LC) maturation (PMID: 19454665), lacking dendritic epidermal T cells (PMID: 21844385). Selective AhR knockout in the epidermal LC causes the loss of LC and immune phenotype skewing (PMID: 32461368). The epidermal keratinocyte-specific constitutive AhR activation leads to atopic dermatitis (AD)-like inflammation (PMID: 27869817), while AhR agonist 6-formylindolo[3,2-b]carbazole attenuates imiquimod-induced psoriasis (PS)-like dermatitis (PMID: 24909886). The authors should modify the chapters accordingly with the possible addition of corresponding tables for better readability.

2. Chapter 7

   Tapinarof looks like one of the most promising topical AhR agonists therapeutically applied to treat AD/PS, and results from clinical trials are already published. The authors should review the progress of clinical trials (see ClinicalTrials.gov).

3. The quality of English is good, but the reviewer found a minor issue.

  L 248: There were confirmed that; this seems not grammatically correct.

Author Response

[Reviewer 3]

Kim et al. reviewed the molecular mechanism of the aryl hydrocarbon receptor (AhR) for skin immunity.  

After briefly describing the modes of AhR pathway activation, i.e., canonical or non-canonical, they reviewed current understanding regarding the consequences of the activation, which can be either beneficial or detrimental. They also reviewed AhR ligands, either exogenous or endogenous, with disease-modifying effects.

Although this manuscript is well-written, the reviewer raises some concerns.

  1. Chapters 3 and 4

 The reviewer strongly agrees with the manuscript arrangement since AhR activation can be therapeutic or disease-causing. However, some important studies regarding the roles of AhR in vivo are missing. For instance, AhR-deficient mice have defective retinoic acid metabolism (PMID: 9230184), resistant to benzo[a]pyrene-induced carcinogenesis (PMID: 10639156), defective Langerhans cell (LC) maturation (PMID: 19454665), lacking dendritic epidermal T cells (PMID: 21844385). Selective AhR knockout in the epidermal LC causes the loss of LC and immune phenotype skewing (PMID: 32461368). The epidermal keratinocyte-specific constitutive AhR activation leads to atopic dermatitis (AD)-like inflammation (PMID: 27869817), while AhR agonist 6-formylindolo[3,2b]carbazole attenuates imiquimod-induced psoriasis (PS)-like dermatitis (PMID: 24909886). The authors should modify the chapters accordingly with the possible addition of corresponding tables for better readability.

We would like to thank the Reviewer for the comment. Each was additionally inserted.

Mouse experiments have not been added as the arrangement of the figures describes human disease. Please review again.

  1. Chapter 7

 Tapinarof looks like one of the most promising topical AhR agonists therapeutically applied to treat AD/PS, and results from clinical trials are already published. The authors should review the progress of clinical trials (see ClinicalTrials.gov).

We would like to thank the Reviewer for the comment. I checked again. The main document has been modified.

  1. The quality of English is good, but the reviewer found a minor issue.

 L 248: There were confirmed that; this seems not grammatically correct.

We would like to thank the Reviewer for the comment. Edited again.

It was confirmed that AhR expression is increased in patients with chronic inflammatory skin disease.

Round 2

Reviewer 3 Report

Thank you very much for the revision. Although some points were addressed appropriately, this manuscript needs improvement for publication. It is crucial to refer to previously published studies in chapters 3 and 4.

1. L118, L184–L197: no reference was made to each published material. The author should correct the reference accordingly.

2. L198–201: [45] does not refer to the original research article written by Rico-Leo at el.; it is a review article. 

3. L147: Corneocytes are terminally differentiated, ‘mummified’ keratinocytes that do not possess the differentiation potential.

Author Response

1. L118, L184–L197: no reference was made to each published material. The author should correct the reference accordingly.

Thank you so much for the good point.

All references are attached. Changes in references are indicated in green, and fluorescence of references is newly attached.

2. L198–201: [45] does not refer to the original research article written by Rico-Leo at el.; it is a review article. 

 The reference is incorrect and has been re-attached.

3. L147: Corneocytes are terminally differentiated, ‘mummified’ keratinocytes that do not possess the differentiation potential.

The part where there was a misunderstanding was deleted.